# Growth Profiles of Children and Adolescents Living with and without Perinatal HIV Infection in Southern Africa: A Secondary Analysis of Cohort Data

**DOI:** 10.3390/nu15214589

**Published:** 2023-10-28

**Authors:** Andrea M. Rehman, Isaac Sekitoleko, Ruramayi Rukuni, Emily L. Webb, Grace McHugh, Tsitsi Bandason, Brewster Moyo, Lucky Gift Ngwira, Cynthia Mukwasi-Kahari, Celia L. Gregson, Victoria Simms, Suzanne Filteau, Rashida A. Ferrand

**Affiliations:** 1MRC International Statistics and Epidemiology Group, London School of Hygiene & Tropical Medicine, London WC1E 7HT, UKemily.webb@lshtm.ac.uk (E.L.W.); victoria.simms@lshtm.ac.uk (V.S.); 2MRC/UVRI and LSHTM Uganda Research Unit, Entebbe P.O. Box 49, Uganda; isaac.sekitoleko@mrcuganda.org; 3Department of Clinical Research, London School of Hygiene & Tropical Medicine, London WC1E 7HT, UK; 4The Health Research Unit Zimbabwe, Biomedical Research and Training Institute, Harare P.O. Box A178, Zimbabwevakoma@gmail.com (T.B.); cynthia.kahari@lshtm.ac.uk (C.M.-K.); 5Malawi-Liverpool-Wellcome Trust Clinical Research Programme, Blantyre 312233, Malawi; brewmoyo101@gmail.com (B.M.); lucky-gift.ngwira@lstmed.ac.uk (L.G.N.); 6Health Economics Policy Unit, Kamuzu University of Health Sciences, Blantyre 312225, Malawi; 7Department of Infectious Disease Epidemiology, London School of Hygiene & Tropical Medicine, London WC1E 7HT, UK; 8Musculoskeletal Research Unit, Translational Health Sciences, Bristol Medical School, University of Bristol, Bristol BS8 1QU, UK; celia.gregson@bristol.ac.uk; 9SAMRC/Wits Developmental Pathways for Health Research Unit, School of Clinical Medicine, University of the Witwatersrand, Johannesburg 2050, South Africa; 10Department of Population Health, London School of Hygiene & Tropical Medicine, London WC1E 7HT, UK; suzanne.filteau@lshtm.ac.uk

**Keywords:** adolescents, growth profile, longitudinal, HIV, latent class analysis, Zimbabwe, Malawi

## Abstract

Impaired linear growth and slower pubertal growth can be associated with perinatal HIV infection. We characterised growth relative to population norms, among the full adolescent period in southern Africa to better understand processes leading to morbidity in adulthood. We conducted a secondary analysis of 945 adolescents aged 8–20 years from urban Malawi and Zimbabwe; we included children with HIV (CWH), an uninfected comparison group from a cohort study, and CWH with co-morbid chronic lung disease (CLD) from a randomised controlled trial. We used latent class analysis of anthropometric Z-scores generated from British 1990 reference equations at two annual time-points, to identify growth trajectory profiles and used multinomial logistic regression to identify factors associated with growth profiles. Growth faltering (one or more of weight-for-age, height-for-age, or BMI-for-age Z-scores < −2) occurred in 38% (116/303) of CWH from the cohort study, 62% (209/336) of CWH with CLD, and 14% (44/306) of HIV-uninfected participants. We identified seven different growth profiles, defined, relatively, as (1) average growth, (2) tall not thin, (3) short not thin, (4) stunted not thin, (5) thin not stunted, (6) thin and stunted and (7) very thin and stunted. Females in profile 3 exhibited the highest body fat percentage, which increased over 1 year. Males at older age and CWH especially those with CLD were more likely to fall into growth profiles 4–7. Improvements in height-for-age Z-scores were observed in profiles 6–7 over 1 year. Interventions to target those with the worst growth faltering and longer-term follow-up to assess the impact on adult health are warranted.

## 1. Introduction

Growth faltering, including impaired childhood linear growth and pubertal delay, is a common co-morbidity among adolescents with perinatally acquired HIV [1]. HIV can impair the adolescent growth spurt, including delaying its onset and reducing linear growth velocity [2]. There is increasing recognition that despite antiretroviral therapy (ART), children growing up with HIV in low- and middle-income countries (LMICs) are at risk of multisystem comorbidities which may further impair growth and body composition, which in turn increase the risk of adult chronic non-communicable disease (NCD). Undernutrition, with or without HIV infection, can further impact the timing of the growth spurt; recent results published from the Birth to 20 cohort in South Africa (HIV uninfected children) showed that children beginning puberty stunted were more likely to reach adulthood shorter and with lower fat-free mass, primarily due to the effects of reduced growth velocity during puberty [3]. Research is ongoing as to the effect of early life undernutrition on the risk of adult NCD [4].

Adolescence offers an opportunity for catch-up growth [5] to mitigate the effects of early life growth faltering within the life course. Before proposing specific interventions to optimise growth, such as nutritional supplementation, education, or behaviour change, a greater understanding is needed of how adolescents living with HIV are affected by growth faltering in comparison to their uninfected peers. Some studies have reported on growth among school-aged children living with HIV [6,7,8,9,10,11], but studies of older adolescents, which include an HIV-negative comparison group, and particularly from sub-Saharan Africa, are scarce. The purpose of the current study was to characterise growth relative to population norms among adolescents in southern Africa. We include three groups aged 8–19 years: (1) those without HIV, (2) those living with HIV, stable on ART [12], and (3) those living with HIV, on ART, and living with chronic lung disease, a common comorbidity associated with HIV infection in Africa [13].

## 2. Materials and Methods

### 2.1. Study Population

This study was a secondary data analysis of prospectively collected anthropometry data, at one or two time points for each participant, from two studies, one cohort and one randomised controlled trial (RCT). Children living with HIV (CWH) and HIV-negative individuals were recruited into the cohort study (IMVASK) [12]. In addition, CWH who had documented chronic lung disease (CLD) were recruited into an RCT (BREATHE) [13]. For this analysis, we excluded 11 RCT participants who did not meet inclusion criteria for CLD. All participants included in this secondary data analysis were residents within the peri-urban public hospital catchment areas in Harare, Zimbabwe or Blantyre, Malawi, and data were collected between 2016 and 2021.

The IMpact of Vertical HIV infection on child and Adolescent SKeletal development (IMVASK) cohort study, conducted in Zimbabwe between August 2018 and January 2021, enrolled 8–16-year-olds, using quota-based sampling, stratified by age group (8–10, 11–13, and 14–16 years), sex, and HIV status, aiming to recruit 50 in each stratum [12]. Perinatally infected CWH who had been on ART for at least two years, and with knowledge of HIV status (for those over 11 years), were recruited from the outpatient clinics at the public-sector hospitals, Parirenyatwa and Harare Central (now Sally Mugabe) Hospitals. Participants not living with HIV were recruited to be representative of the Harare population, from 6 (randomly selected from 153) government schools, 3 primary and 3 secondary, with participants selected using probability proportional to school roll and quota-based sampling. One exclusion criterion was HIV infection identified at screening at the study clinic, with referral to HIV care as appropriate. Participants were studied at two visits, one year apart. The study is registered at ISRCTN12266984.

The Bronchopulmonary Function in Response to Azithromycin Treatment for Chronic Lung Disease in HIV-infected Children (BREATHE) randomised controlled trial was an individually randomised, multi-site, placebo-controlled, double-blind trial of azithromycin versus placebo, with a 1:1 allocation ratio stratified by site, and was conducted between June 2016 and August 2019 [13]. There were 347 6–19-year-olds enrolled and randomised over the two sites in Zimbabwe and Malawi. The Zimbabwe participants were recruited from the same hospitals as IMVASK participants. Participants were living with co-morbid HIV and CLD, defined as a forced expiratory volume in 1 s (FEV1) Z-score of <−1 with lack of reversibility (<12% improvement with salbutamol 200 μg inhaled using a spacer).

Trial eligibility criteria were perinatal HIV transmission, established on ART (minimum 6 months), intention to remain in the area, knowledge of HIV status (for those over 11 years), and meeting criteria for CLD. Exclusion criteria included pregnancy, creatinine clearance < 30 mls/minute, alanine aminotransferase (ALT) >2 times the upper limit of normal, evidence of prolonged QTc interval, and active lung infection (acute respiratory exacerbation or tuberculosis for which all participants were screened actively at baseline). Participants, health care providers, and outcome assessors were blinded to trial allocation. The trial intervention had no effect on anthropometry over 48 weeks [14]. Follow-up visits were at enrolment (week 0), then at 2-, 12-, 24-, 36-, 48- and, for a subset, 60- and 72-weeks. For this secondary data analysis, data from week 0 and 48 were used in our analysis, and data from the other visits were used for data cleaning. The trial is registered at ClinicalTrials.gov Identifier: NCT02426112.

### 2.2. Data Collection

IMVASK participants had duplicate (or triplicate in discrepant cases) anthropometry measurements, while BREATHE participants had single measurements. Height was measured to the nearest 0.1 cm and weight to the nearest 0.1 kg, with the mean value used for analysis [15]. Measurements were converted to anthropometric Z-scores using the 1990 British reference equations to generate weight-for-age (WA), height-for-age (HA), and Body Mass Index-for-age (BMI) Z-scores [16]. We defined overweight as a BMI Z-score > 1. Underweight, stunting, and thinness were defined as Z-scores < −2 for WA, HA and BMI.

GeneXpert^TM^ HIV-1 Viral Load (Cepheid, Sunnyvale, CA, USA) was used to determine HIV viral load, with a lower limit of detection of 40 copies/mL and CD4 cell count/µL was measured using an Alere PIMA CD4 machine (Waltham, Massachusetts, USA) in both studies.

Diet and body composition data were collected in the IMVASK cohort study. Food frequency, focused on protein and calcium-rich foods, was captured using a validated tool adapted for Zimbabwe [17]. Body composition, including fat mass and fat-free mass as a proxy for lean mass, was obtained from dual energy X-ray absorptiometry (DXA) total body scans under standard conditions using a Hologic QDR Wi densitometer (Hologic Inc., Bedford, MA, USA) with Apex Version 4.5 software for scan analysis. Given the inherent sex differences in body composition, we present sex-stratified data. Height and weight were checked for consistency over time and against DXA-measured total body weight; implausible values were removed from analysis if DXA-measured values were not available. BREATHE study height and weight were cleaned prior to publication of study outcomes, and no further checks were undertaken in the present study.

Participant data were captured on Google NexusTM tablets, (Google, Mountain View, CA, USA) using OpenDataKit Aggregate v1.4 software and uploaded to Microsoft Access databases (Microsoft, Redmond, WA, USA). Data were merged and analysed using Stata v15 or v16.1 software (StataCorp, Texas, USA).

### 2.3. Statistical Methods

We used latent class structural equation models, the STATA gsem command, including WA, HA and BMI Z-scores at enrolment and one-year (two time points were available for most individuals—Figure 1) to identify profiles of growth trajectories. To determine the optimal number of profiles which fitted the data best, we used the Bayesian information criteria (BIC, lower values are better), entropy values (values closer to 1 are better), predicted posterior probabilities (considered the certainty that a given participant was a member in their particular allocated profile), and ensured sample sizes in each group were a minimum of 40 individuals (5% of the sample size) [18]. We did not conduct formal sample size calculations for this secondary data analysis, but our fixed sample size of 945 individuals meets recommendations of at least 20 participants per explanatory variable in order to have adequate statistical power when fitting structural equation models [19]. We explored associations between risk factors and growth profiles using multinomial logistic regression to generate relative risk ratios (RRR) and 95% confidence intervals (CI). Available risk factors were sex [20], age [3], HIV and comorbidity status [9,21,22], and country of residence [23], and for CWH [8], age at initiation of ART, HIV viral load and CD4 cell count.

## 3. Results

### 3.1. Characteristics of the Study Population

The analysis included 945 individuals aged 6–19 years at baseline, 639 (68%) of whom were living with HIV, 472 (50%) of whom were female, with mean age 13.4 (SD 3.0) years (Figure 1, Table 1). CWH with CLD in BREATHE (336/945 participants) were older than IMVASK participants; 28% of CWH with CLD were Malawian; and (as expected) follow-up time between visits was shorter for BREATHE than IMVASK participants. CWH in the IMVASK cohort (303/945 participants) commenced ART earlier in life than those from the BREATHE RCT; had better-controlled HIV, with higher CD4 cell count; and a greater proportion were virally suppressed (Table 1). The follow-up visit was conducted a median of 53 (interquartile range 48, 62) weeks after enrolment for 798 (84%) participants who contributed two time points to the analysis; 147 participants contributed only the baseline measurement to analysis.

At enrolment, anthropometric Z-scores were lower among CWH from the BREATHE RCT than CWH from the IMVASK cohort and highest among HIV-negative children. Mean Z-scores were higher for females than males in all of these categories. Over one year of follow-up, among those with two measurements, there was no change in BMI Z-Score (mean difference 0.02 higher, 95% CI 0.06 higher to 0.02 lower, *p*-value 0.39, *n* = 796), or WA Z-Score (mean difference 0.02 higher, 95% CI 0.05 higher to 0.01 lower *p*-value 0.28, *n* = 796), with a small improvement in HA Z-Score (mean difference 0.04 higher, 95% CI 0.06 higher to 0.02 higher, *p*-value 0.0009, *n* = 798).

Growth faltering (one or more of stunting, underweight, or thinness) at enrolment was more common among CWH from BREATHE (209/336, 62%) than among the IMVASK cohort, CWH (116/303, 38%), and HIV-negative individuals (44/306, 14%). Among CWH in the IMVASK cohort growth faltering was also more prevalent in older children (30/58, 52% aged ≥ 15) compared to younger children (33/99, 33% aged 6 to 10 years), chi2 *p*-value = 0.06. Males were more likely to have growth faltering than females (46% versus 32%, chi2 *p*-value < 0.001).

A small proportion of children were classified as overweight, many of whom were also stunted: 36/305 (12%) HIV-negative individuals with mean HA Z-Score −0.15 (range −1.9, +2.05), 9/301 (3%) CWH from IMVASK cohort with mean HA Z-Score −1.42 (range −2.94, +0.26), and 12/336 (4%) CHW from BREATHE with mean HA Z-Score −1.66 (range −2.73, −0.63). In total, only 6/942 individuals were stunted and overweight.

### 3.2. Identification and Summary of Growth Profiles

Latent class analysis revealed a seven-class model which best explained weight-for-age, height-for-age and BMI-for-age trajectories, producing the closest entropy value to 1 where all classes contained at least 5% of the study population. The class sizes ranged from 50 (5.3%) to 265 (28.0%) individuals (Table 2). Z-scores are based compared to the 1990 UK reference. The seven class profiles (Figure 2, Figure 3 and Figure 4) can be summarised as by the following profiles:

Average-growth individuals had BMI, weight, and height closest to the average reference values, although WA and HA Z-scores were consistently around 0.5 SD below the reference. Z-scores were stable over the year of follow-up. The majority of HIV-negative individuals were in this profile. This was the profile with the largest number of individuals (28%).

Taller (than other profiles) not thin individuals (7% of the sample) whose weight and BMI were around 1 SD above the reference, higher weight for their height, and HA Z-score was around 0.5 above the reference. Few were obese; five individuals had weight-for-age Z-score > 2 and eight individuals had BMI-for-age Z-score > 2. Height-for-age Z-scores of individuals in this profile reduced over follow-up whilst BMI Z-scores slightly increased. This profile was the second most common for school children but was rare among CWH. Most individuals were older females. Males in this profile had the highest total body fat percentage of all the profiles.

Short not thin individuals (7% of the sample) whose BMI was around 1 SD above reference values and HA were around −1.5 SD below reference values. On average, at follow-up, height-for-age Z-scores fell whilst weight-for-age Z-scores remained stable in these individuals. This profile was more common in older females and in CWH. Not only did these individuals have higher weight in proportion to their height, the females in this profile had the highest total body fat percentage of all the profiles. There were no individuals with weight-for-age Z-score > 2 and only two individuals with BMI-for-age Z-score > 2.

Stunted but not thin individuals with BMI around 0.5 SD below reference values who were both stunted and had lower weight. Weight-for-age Z-scores were stable over time, while height-for-age Z-scores increased slightly. This was the most common profile for CWH and could indicate a legacy of early growth faltering; this profile was equally common among males and females. This was the second largest profile, comprising 26% of individuals.

Thin but not stunted individuals (11% of the sample) with BMI around 1.9 SD below reference values who had low weight but who were not stunted. All three anthropometry Z-scores increased slightly over follow-up among these individuals. This profile was more common among children aged 11–14 years at baseline and the trajectory could indicate a recent growth spurt and pubertal transition.

Thin and stunted individuals were around 3 SD below reference values for both weight and height. There was some evidence that height-for-age Z-scores increased over time, but both weight-for-age and BMI-for-age Z-scores decreased over time, and there was heterogeneity in change over time. This profile was more common among CWH, males and older adolescents, and anthropometry indicated growth faltering. This profile was reasonably common, containing 16% of individuals.

Very thin and stunted individuals had weight and height less than 4 SD below reference values at baseline. On average, these individuals’ Z-scores improved during follow-up, but there was greater heterogeneity in growth than for other profiles. Similar to profile 6, this profile was more common among males of older ages and indicated growth faltering. The majority of individuals in this profile had HIV with CLD. This was the rarest profile including 5% of individuals.

### 3.3. Factors Associated with Growth Profiles

Sex (Wald *p*-value < 0.0001), HIV status (Wald *p*-value < 0.0001) and age (Wald *p*-value = 0.0003) were all associated with latent class growth profile (Table 3), but country of residence was not (Wald *p*-value = 0.33). Females were more likely to exhibit profile 1 (average growth), profile 2 (tall not thin), or profile 3 (short not thin), and males were more likely to be thin and/or stunted profiles 4–7. Profiles where stunting was common (3, 4, 6 and 7), were associated with older age and HIV-infection, particularly profile 7, very thin and stunted. Compared to those who were HIV-negative, the RRR of being in profile 6 (thin and stunted) and 7 (very thin and stunted) was greatest for those CWH with comorbid CLD (from the BREATHE RCT). We lacked power to explore HIV-related factors associated with growth profiles because of low numbers in some profiles, and did not find evidence for association of growth profile with age at ART initiation (*p*-value = 0.95), viral suppression (*p*-value = 0.24) or CD4 cell count (*p*-value = 0.27, Appendix A).

Among IMVASK participants (for whom DXA scans were taken) at enrolment, mean percentage total body fat (TBF) among females in profile 1 was 26.8 (SD 4.1) and among males was 20.9 (SD 3.2). profiles 3–6 had lower % TBF, and profiles 2 (tall not thin), and 3 (short not thin), had higher % TBF (Figure 4). Differences between profiles in trunk fat were materially similar (Appendix A). There were some differences in the consumption of dairy, eggs and fish among profile groups, with a greater proportion of those in profiles 1, 2 and 3 consuming those foods more than 1–2 days per week compared to those in profiles 4–7. Meat consumption more than 2 days per week was almost universal (Appendix A) so we were unable to statistically test for an association with growth profile.

## 4. Discussion

This secondary data analysis of cohort and RCT data characterised seven growth profiles among children and adolescents from Southern Africa and included individuals living with HIV, with both HIV and chronic lung disease, as well as HIV-negative children. The majority of individuals exhibiting average growth (profile 1), defined as Z-scores around 0.5 SD below the UK population reference, were HIV-negative school children. Two profiles differentiated individuals who had BMI-for-age Z-scores around 1 SD above the UK reference, with their weight relatively high for their height, into those who were tall for their age (profile 2, more likely to be older HIV-negative children) and those who were short for their age (profile 3, more likely to be older CWH). Although few of these individuals were classified as obese, the highest percentage total body fat coincided with these profiles which were dominated by females. The second most common profile was stunted not thin (profile 4), which were a group of children, the majority CWH, whose body size was proportionate to their shorter height. Some of these individuals were younger than those in profiles 2 and 3, so they might gain further height as they progress through puberty, but have started puberty with a legacy of early growth faltering. Profile 5, thin not stunted, was more common among 11–14 year-olds and could indicate a recent growth spurt and pubertal transition. Profile 6, thin and stunted and profile 7: very thin and stunted were more common among CWH, males and at older age, and indicate growth faltering; profile 6 was the largest profile among CWH with CLD (BREATHE RCT). profile 7 was the rarest profile, with the lowest Z-scores but the vast majority in this profile were CWH with co-morbid CLD from the BREATHE RCT.

HIV was associated with growth profiles demonstrating poor linear growth (profiles 3–7) supporting other studies which have shown growth faltering in early childhood among CWH [1,24] and our findings expand this knowledge to the later adolescent period. We did not observe catch-up in linear growth over one year of follow-up, that is, Z-scores moving closer to average population norms, except for profiles 6 and 7 who were thin and stunted at enrolment. In spite of improvements in height-for-age Z-score over follow-up, children in these profiles remained stunted and thin. Overall, we observed little change in Z-scores over one-year of follow-up (in profiles 1–5), indicating that early growth trajectories may continue to have persistent effects through adolescence, a finding consistent with the South African Birth-to-20 s cohort, where greater linear growth up to five years of age was associated with greater subsequent adolescent growth [20]. CWH with co-morbid CLD from BREATHE were more likely to have the most malnourished (growth profiles 6 and 7). Among the very thin and stunted, profile 7, we observed mean increases in HA Z-score of 0.32 SD and in BMI Z-score of 0.19 SD indicating the potential for some improvements (from very low baseline Z-scores), although this could be explained by regression towards the mean. For these children, mean Z-scores remained very low at follow-up, and given they were older than children in other profiles the possibility for catch-up growth appeared largely missed, unless progression of pubertal transition was very slow. It would be important to follow-up these individuals into adulthood to determine if the pubertal transition was prolonged. Similarly, among those in profile 6 (thin and stunted), smaller increases in HA Z-score of 0.14 suggested some possible improvement over time. Ideally, potential interventions would prevent children developing the most malnourished profiles (6 and 7).

There is strong evidence that adults with higher body fat are at increased risk of type 2 diabetes and cardiovascular disease [25]. We identified a growth profile for 69 individuals, whom we classified as short not thin (profile 3), who were older and more likely to have HIV. The females in this profile had both the highest percentage total body fat (and trunk fat) and the largest one-year increase in body fat among the profiles. Higher weight gain up to the age of two years was associated with faster body fat gains in adolescence in black South Africans (without HIV) particularly in females [20], and there is evidence from Zambia that the highest body fat in early adolescence was amongst those with adequate growth up to the age of three [26]. More research on African cohorts is needed into the long-term effects of experiencing early and persistent linear growth faltering, combined with life-long ART and its effects on body composition during adolescence (and risk of chronic disease in adulthood).

Individuals in profiles 6 and 7, those thin and stunted, from the IMVASK cohort (where dietary intake was recorded) tended to report lower dairy consumption than other profiles. It is not possible to ascertain a causal link between lower dairy consumption and lower anthropometric Z-scores in this dataset because growth and diet have been collected contemporaneously. However, given that dairy consumption is associated with linear growth in adolescence [27], we might postulate that lower dairy consumption habits in these two profiles may have contributed to their lower linear growth prior to enrolment in the study. The severity of stunting in these profiles warrants follow-up of these individuals to assess for increased morbidity and mortality, reduced physical and cognitive capacity including poorer educational performance, and an elevated risk of metabolic disease into adulthood [28,29,30,31].

Our study has limitations. We used the UK 1990 reference equations, which might be considered out-of-date compared to the WHO reference, although they do provide WA Z-scores for those aged 0 to 23 (allowing the inclusion of 20-year-olds at follow-up), where the WHO has an upper limit of age 10. However, the choice of reference is not critical for our aim to compare between groups within our population, and we encompassed the full range of growth faltering including WA, HA and BMI together using the British standards. Our sample has over-representation of CWH with comorbidity (which could have included some enrolled into the IMVASK study where FEV1 was not measured), so we were unable to provide estimates of the population prevalence of our growth profiles. On the other hand, because those with chronic conditions were over-represented, we were able to differentiate between two profiles which both constitute those thin and stunted, given the high prevalence of growth faltering among CWH with co-morbid CLD from the BREATHE RCT. We had 16% loss to follow-up overall and this was related to factors which are also associated with growth profile [14]. However, given the small changes in Z-scores over time, this would not have a major impact on our findings even if dropout was related to nutritional status (a missing not-at-random assumption). The short length of follow-up provided limited scope for investigating longer-term changes over time (dependant on whether follow-up coincided with peak growth velocity) and limited full exploration of whether catch-up growth might occur if the pubertal transition is slower for those with HIV—particularly for males, who transition later than females; further follow-up of the cohort could provide greater insights. Assignment to latent class is dependent on predicted probabilities, which means that true class assignment is not certain; we were unable to validate our latent classes on another dataset, and so the profiles have potential for misclassification bias. Future work could explore and validate the profiles that we identified on other datasets from sub-Saharan Africa. For simplicity, we have given to each latent class names which describe the main characteristics of the individuals in that class. Finally, because we included studies which recruited from the same catchment area, it is possible that a small number of individuals were included in both studies in this dataset.

The wide age range of our study, longitudinal nature, robust statistical methods, and ability to compare three groups (HIV-negative children, CWH, and CWH with co-morbid CLD), the majority of whom are from the same population, are strengths of our study. We had a limited number of variables which were common across studies and so were constrained in our ability to identify factors associated with growth profiles. However, we provide a framework for exploring growth faltering among adolescents and have identified the most vulnerable in terms of those having weight and height 3 SD below UK population norms.

## 5. Conclusions

We revealed seven growth profiles over one year of follow-up. CWH and co-morbid CLD had the greatest degree of growth faltering, and this was most common among older males. Further work is ongoing to identify the effects of HIV on skeletal maturation and to explore the potential for supplementation to improve growth in adolescents living with HIV. Longer term follow-up of the most malnourished is warranted to assess the impact on adult health. Longer term follow-up of those with the highest body fat percentage at the end of adolescence, and those who had a high weight for their height, is also warranted to assess the risk of developing diabetes and cardiovascular disease.

## Figures and Tables

**Figure 1 nutrients-15-04589-f001:**
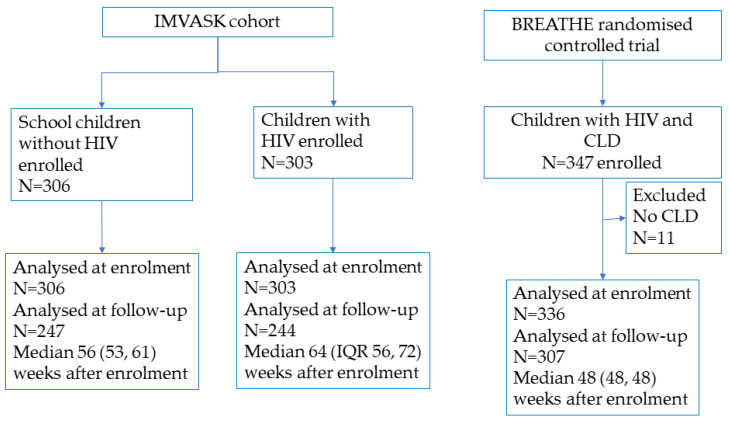
Participant flow diagram.

**Figure 2 nutrients-15-04589-f002:**
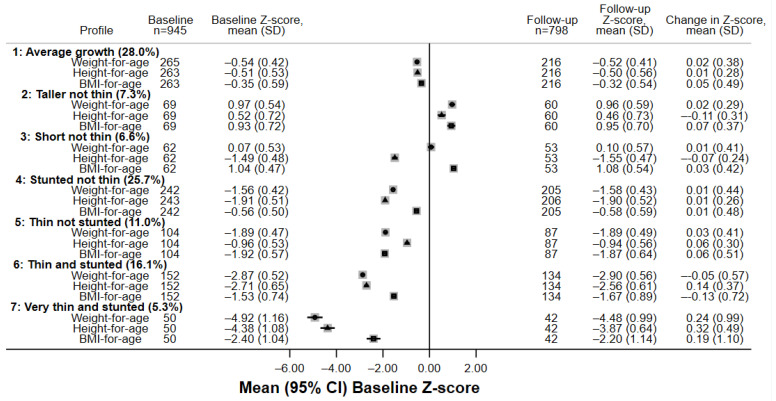
Participant characteristics and anthropometry for each latent class profile *n* = 945. WAZ—weight-for-age Z-score (solid circles), HAZ—height-for-age Z-score (solid triangles), BMIZ—body mass index-for-age Z-score (solid squares).Mean and SD are presented for age.

**Figure 3 nutrients-15-04589-f003:**
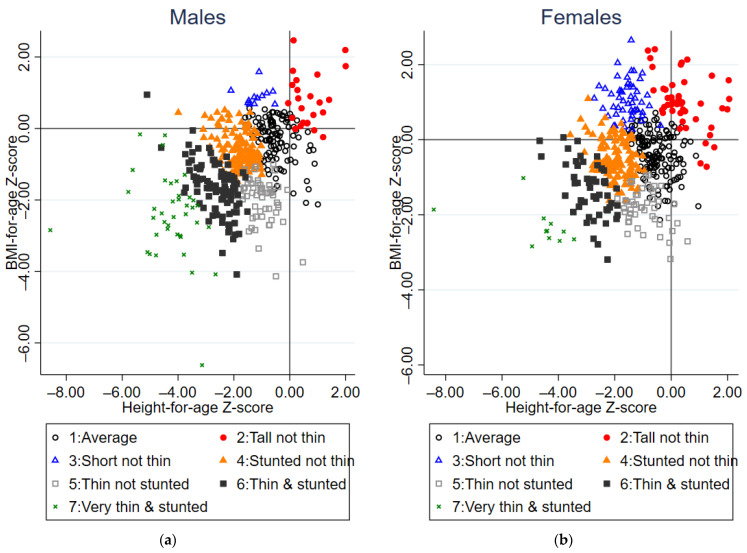
Scatterplot of BMI and height-for-age Z-scores at baseline by latent class group for (**a**) Males, *n* = 473 and (**b**) Females, *n* = 472.

**Figure 4 nutrients-15-04589-f004:**
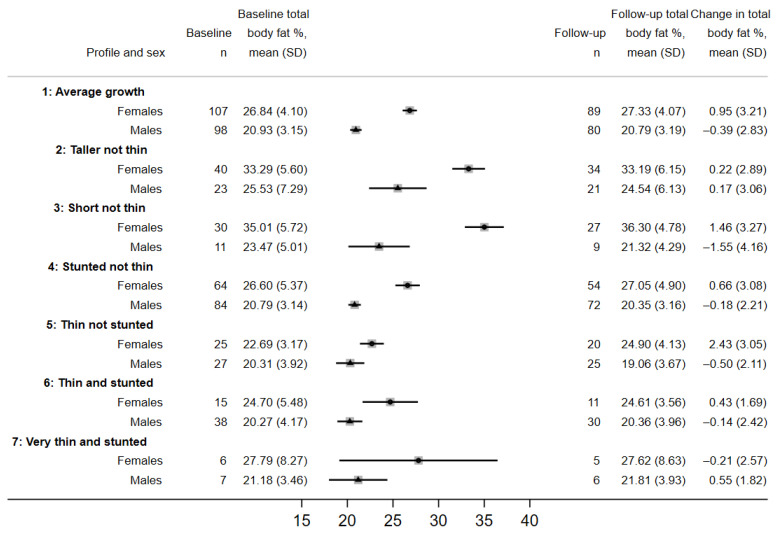
Total body fat percentage for each latent class profile by females (solid circles) and males (solid triangles) among *n* = 609 IMVASK participants. Body fat percentage available for *n* = 575 at baseline and *n* = 483 at follow-up.

**Table 1 nutrients-15-04589-t001:** Participant characteristics at enrolment, by HIV and co-morbidity status.

Characteristics *	HIV− IMVASK Cohort	HIV+ IMVASK Cohort	HIV+ and CLD+ BREATHE RCT
*n*	306	303	336
Mean age, years (SD)	12.5 (2.5)	12.4 (2.5)	15.0 (3.2)
Female sex, *n* (%)	155 (50.7)	151 (49.8)	166 (49.4)
Zimbabwean, *n* (%)	306 (100)	303 (100)	241 (71.7)
Malawian, *n* (%)	-	-	95 (28.3)
Median age at ART initiation, years (IQR)	-	3.8 (1.8, 6.9)	8.5 (5.7, 11.6)
HIV viral load <1000 copies/mL, *n* (%)	-	212 (79.1)	187 (56.0)
CD4 cell count ≥500 cells/µL, *n* (%)	-	230 (79.9)	208 (61.9)
Mean weight (SD), kg	40.2 (12.2)	34.0 (9.4)	38.5 (11.1)
Mean Weight-for-age Z-score (SD)	−0.50 (1.09)	−1.47 (1.19)	−2.17 (1.46)
Underweight (WA Z-score < −2), *n* (%)	23 (7.5)	80 (26.4)	176 (52.4)
Mean height (SD), cm	147.5 (13.5)	140.0 (12.8)	146.0 (14.1)
Mean Height-for-age Z-score (SD)	−0.59 (1.02)	−1.65 (1.12)	−2.11 (1.22)
Stunted (HA Z-score < −2), *n* (%)	22 (7.2)	95 (31.6)	168 (50.0)
Mean BMI (SD)	18.0 (3.1)	17.0 (2.2)	17.6 (2.8)
Mean BMI Z-score (SD)	−0.27 (1.09)	−0.68 (0.94)	−1.09 (1.15)
Thin (BMI Z-score < −2), *n* (%)	16 (5.3)	28 (9.3)	67 (19.9)
Overweight (BMI Z-Score > 1), *n* (%)	36 (11.8)	9 (3.0)	12 (3.6)

* Missing values: weight (1 HIV− IMVASK cohort), height and BMI (2 HIV+ IMVASK cohort); HIV viral load (35—IMVASK cohort; 2 BREATHE trial); CD4 cell count (15—IMVASK cohort).

**Table 2 nutrients-15-04589-t002:** Latent class model fit and diagnostic criteria.

Number of Classes	Degrees of Freedom	Log Likelihood	Bayesian Information Criteria	Entropy	Smallest Class Size, *n* (%)	Mean Class Posterior Probability
1	12	−8634	17,350	-	945 (100%)	-
2	19	−7587	15,304	0.832	402 (42.5%)	0.95
3	26	−7000	14,178	0.834	116 (12.3%)	0.94
4	33	−6655	13,537	0.907	81 (8.6%)	0.93
5	40	−6433	13,140	0.936	32 (3.4%)	0.93
6	47	−6292	12,905	0.912	48 (5.1%)	0.91
7	54	−6147	12,663	0.946	50 (5.3%)	0.91
8	61	−6003	12,425	0.955	38 (4.0%)	0.91

The 7-class model was determined to fit the data best. Although the entropy value was slightly decreased compared to the 8-class model, each latent class had at least 40 individuals (5% of the data); the mean posterior probability predicting class membership was comfortably above 0.80 [18].

**Table 3 nutrients-15-04589-t003:** Multinomial logistic multivariable regression showing factors independently associated with growth profiles among the full sample, *n* = 945.

Factors	Profile 2: Tall not Thin	Profile 3: Short not Thin	Profile 4: Stunted not Thin	Profile 5: Thin not Stunted	Profile 6: Thin and Stunted	Profile 7: Very Thin and Stunted
	RRR (95% CI)	RRR (95% CI)	RRR (95% CI)	RRR (95% CI)	RRR (95% CI)	RRR (95% CI)
**Sex**						
Male	Ref.	Ref.	Ref.	Ref.	Ref.	Ref.
Female	1.70 (0.97, 3.00)	2.86 (1.44, 5.68)	0.58 (0.40, 0.84)	0.56 (0.35, 0.90)	0.30 (0.19, 0.48)	0.15 (0.07, 0.31)
**Age, years**						
6 to 10	1.58 (0.71, 3.55)	0.22 (0.09, 0.54)	0.78 (0.47, 1.29)	0.55 (0.28, 1.10)	0.65 (0.35, 1.21)	0.06 (0.01, 0.29)
11 to 14	1.29 (0.59, 2.81)	0.43 (0.22, 0.83)	0.69 (0.44, 1.10)	0.96 (0.54, 1.70)	0.76 (0.44, 1.30)	0.30 (0.14, 0.64)
15 to 19	Ref.	Ref.	Ref.	Ref.	Ref.	Ref.
**HIV status**						
HIV−	Ref.	Ref.	Ref.	Ref.	Ref.	Ref.
HIV+	0.18 (0.08, 0.41)	1.37 (0.69, 2.73)	3.30 (2.13, 5.10)	1.75 (0.98, 3.15)	8.26 (3.92, 17.4)	28.8 (3.67, 226.8)
HIV+ CLD+	0.20 (0.06, 0.70)	1.64 (0.75, 3.60)	4.02 (2.33, 6.93)	4.46 (2.35, 8.49)	25.5 (11.6, 56.1)	57.3 (7.30, 449.7)
**Country**						
Zimbabwe	Ref.	Ref.	Ref.	Ref.	Ref.	Ref.
Malawi	0.89 (0.08, 9.46)	1.10 (0.32, 3.74)	1.58 (0.70, 3.55)	0.81 (0.30, 2.20)	1.54 (0.68, 3.48)	2.78 (1.01, 7.60)

The reference profile is profile 1: Average growth. RRR—relative risk ratio; CLD—chronic lung disease. RRR are interpreted relative to the reference profile; for example, when comparing females to males, the relative risk ratio of being in profile 3 compared to profile 1 is 2.86 (95%CI 1.44, 5.68) given the other variables in the model are held constant. In other words, females are more likely than males to be in profile 3 compared to profile 1. Overall Wald *p*-values: Sex *p*-value < 0.0001, HIV status *p*-value < 0.0001, age *p*-value = 0.0003, country of residence *p*-value = 0.33.

## Data Availability

The data presented in this study are available on request from the corresponding author. The data are not publicly available yet due to unfinished analysis of the data by PhD students.

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
