# Peer review of "Growth Profiles of Children and Adolescents Living with and without Perinatal HIV Infection in Southern Africa: A Secondary Analysis of Cohort Data"

_nutrients, 2023, doi:10.3390/nu15214589_

Round 1

Reviewer 1 Report

This paper explores potential growth patterns in children and adolescents living with perinatal HIV compared to those who are not living with perinatal HIV in South Africa.  Individual-level data were from a cohort of children aged 8 to 16 years conducted during period August 2018-January 2021 in Zimbabwe, where those with HIV were identified at outpatient clinics and those without age- and sex quota-sampled from school rolls in the same population; and from a randomised controlled trial of azithromycin v placebo conducted during June 2016 and August 2019 among 6-19 year olds with HIV and chronic lung disease were randomised at two sites in Zimbabwe and Malawi.  Anthropometric measurements were taken at 2 visits, a year apart in the cohort study, and measurements at entry and at 48 weeks were used for the RCT; and were converted to Z-scores using the 1990 British reference equations to account for age.  Data were analysed using latent class structural equation models to identify different growth trajectories, and then subjects’ characteristics were explored within the identified trajectories using multinomial logistic regression.  Seven class profiles were identified.

The age range given in the abstract and in the paper are inconsistent- please check.

Findings in the abstract seem general (lines 36-38) and not necessarily specific to HIV+ compared to HIV-.  Please relate the findings in the abstract to those for the HIV+ groups.

Participants were recruited from the same hospitals in Zimbabwe in both studies.  Were there any participants who were recruited into both studies; how did this pooled study ensure that participants were only entered into the analysis once?

Height and weight were compared to DXA-measured total body weight and implausible values were removed if DXA-measured values were not available.  What proportion of subjects had their data excluded from the analysis?  Was there anything specific to these subjects- known or suspected- that could have affected the study findings?  Only the IMVASK cohort conducted DAX scans- what checks were implemented in the BREATH RCT?

In the methods and abstract, it is stated that latent class analyses were conducted on both anthropometric data at enrolment and at 1 year; the results presented seem to be based on LCA using data at enrolment.  Please clarify.

In the results, it is stated that follow-up time between visits was shorter for the BREATHE than the IMVASK study but this is to be expected given the studies’ designs?

Some characteristics from Table 1 are described in the results but descriptions of the anthropometric summaries could be improved upon in the results.  For instance, Z-scores for weight, height and BMI are less than the British reference population in all three groups but are more so in the HIV+ and HIV+/CLD+ groups than the HIV- group.  T-tests comparing means in the HIV+ groups to the HIV- group could be added.  Also, in the abstract and results, it is stated that “Growth faltering (either stunting, underweight or thinness) at enrolment was more 177 common among CWH from BREATHE (209/336, 62%) than among the IMVASK cohort, 178 CWH (116/303, 38%) and HIV-negative individuals (44/306, 14%).”  Please clarify how these numbers were derived as it is difficult to relate them to the presented tables and figures.

Figure 2 shows the seven classes with the average age, percent females, percent HIV- and the change in Z-score from the first to second measurement by weight, height and BMI- for-age.  Please change the percent HIV- to percent HIV+ as this is the focus of the paper.  Consider positioning the column n of subjects in each group as the first column in the graph (i.e. before mean age) and give the percentage of the total in each group on the figure as is described in the text.

The text in Figures 2 and 4 is quite small and the image quality is not the best.

In figure 4, label the x-axis; and as per figure 2, give the percentage of the total in each group alongside the values for n.  The column title n is used twice but for different values; please change the column titles to make clear what is in each column.

Table 2 gives the multinomial logistic regression findings for each profile- profile 1 is not included in the table?

The authors discuss that this is a study of adolescents with only 1 year of follow-up with little catch-up growth being evident.  The assumption is that progression of pubertal transition may be slow- given the sex difference in age at transition, could this have been missed for older females and not yet determined in males, given the age of participants and only one year between measurements?

Potential comorbid conditions for those with higher body fat are discussed; what about comorbid conditions that could occur among those who are thin and stunted?

Reviewer 2 Report

Thank you for the opportunity to review this interesting work. It is important and valuable for the field. However, I need it has to be improved substantially. I suggest the following:

1. Add a data analysis section that describes all the analytical procedures

2. For the LCA model I would like to see various information criteria and likelihood ratio tests that contrast adjacent models. Indices of homogeneity and separation are required. Without them, the concluded latent class solution cannot be judged properly.

3. The authors use entropy for concluding the optimal number of classes. It is well-known that entropy should not be used in the latent class enumeration process.

4. What types of growth models were tested?

5. What was the power for the LCA and growth models?

6. Figures need enlarged fonts to be read properly.

7. I was particularly intrigued by figure 3. When used as a total sample, the relationship is clearly positive as expected. However, when looking at subgroups, some relationships are clearly negative e.g., in "tall and thin" in females. How do the authors view these contradicting findings?

 8. In table 2, which estimates are significant? I assume when they do not cross 1, but the authors need to place * and indicate p-values.

Again, thank you for the opportunity to review this interesting work.

Round 2

Reviewer 1 Report

Thank you for addressing my comments.

Reviewer 2 Report

The revised version addressed my suggestions satisfactorily.

It will be nice to have a native speaker review the paper.